# The Synthetization and Analysis of Dicyclopentadiene and Ethylidene-Norbornene Microcapsule Systems

**DOI:** 10.3390/polym12051052

**Published:** 2020-05-04

**Authors:** Ionut Sebastian Vintila, Horia Iovu, Andreea Alcea, Andreia Cucuruz, Andrei Cristian Mandoc, Bogdan Stefan Vasile

**Affiliations:** 1National Research and Development Institute for Gas Turbines COMOTI, 061126 Bucharest, Romania; andreea.alcea@comoti.ro (A.A.); andrei.mandoc@comoti.ro (A.C.M.); 2Department of Bioresources and Polymer Science, Faculty of Applied Chemistry and Materials Science, University Politehnica of Bucharest, 011061 Bucharest, Romania; horia.iovu@upb.ro; 3Department of Science and Engineering of Oxide Materials and Nanomaterials, Faculty of Applied Chemistry and Materials Science, University Politehnica of Bucharest, 011061 Bucharest, Romania; andreia.cucuruz@upb.ro; 4National Research Centre for Micro and Nanomaterials, Faculty of Applied Chemistry and Materials Science, University POLITEHNICA of Bucharest, 060042 Bucharest, Romania; bogdan.vasile@upb.ro

**Keywords:** polymer composites, self-healing, thermal stability, FEM analysis, dicyclopentadiene, 5-ethylidene-2-norbornene

## Abstract

The activities of this paper were focused on an in-situ fabrication process for producing two self-healing systems containing dicyclopentadiene and 5-ethylidene-2-norbornene monomers encapsulated in a urea-formaldehyde shell and integration methods applied in the epoxy matrix to analyse and compare the influences of their integration into the neat epoxy matrix. The self-healing systems were first synthesized according to a literature review, and subsequently, an optimization process was conducted for the fabrication process. Neat epoxy specimens were fabricated as reference specimens and subjected to flexural tests. Several integration methods for incorporating the self-healing systems into the epoxy resin were investigated. The optimal method presenting the best dispersion of the healing system was achieved by reducing the viscosity of the epoxy matrix with 10 vol % acetone solution, the addition of a microcapsule in the matrix, and homogenization at 60 °C at 100 rpm. Thermal analysis was performed in order to observe the mass loss obtained with an increasing temperature and phase changes for both poly-urea-formaldehyde (PUF)-dicyclopentadiene (DCPD) and melamine-urea-formaldehyde (MUF)-5-ethylidene-2-norbornene (ENB) systems. The thermogravimetric analysis performed for the PUF-DCPD system indicates a total loss of mass in the range of 30–500 °C of 72.604% and for the MUF-ENB system, indicates a total mass loss in the range of 30–500 °C of 74.093%. Three-point bending tests showed higher mechanical properties for PUF-DCPD (80%) than MUF-ENB (40%) compared to the neat epoxy systems. Numerical simulations were performed to obtain a better understanding of the microcapsule behavior when embedded in an epoxy matrix.

## 1. Introduction

As thermoset polymers possess good physical and mechanical properties (good wettability, corrosion resistance, and high strength), are easy to process, and have a low cost of production, they are extensively used in the manufacturing process of fiber reinforced polymer (FRP) composite structures. During their service life, these polymer composite materials are exposed to several mechanical loads that cause distinctive failure mechanisms with respect to their morphology, when compared to metals. As crack formation appears as a result of mechanical loads, the necessity of repairing these materials became an important aspect in the development of such materials. Therefore, several self-healing techniques have emerged and been investigated in order to overcome this limitation [1,2].

Currently, self-healing techniques are rapidly growing, comprising a wide range of self-healing polymer systems. Drawing inspiration from natural healing systems, researchers have designed and developed materials that can cure themselves after a damage event, thus extending their lifetime.

Over the last years, distinct, nature-inspired concepts for delivering an autonomous fluid system to the damaged area have been developed [3,4,5,6], in order to completely repair the FRP composites. With respect to traditional materials, self-healing materials possess the ability to heal themselves when subjected to failure through fracture or fatigue. To date, the majority of studies in the self-healing area have been concentrated on polymers and polymer composites as they are widely used in many important industrial applications (aeronautic, aerospace, marine, transport, military, medicine, etc.). To be considered a self-healing material, a specimen has to possess the following properties:-The ability to heal the material automatically;-The ability to heal the damage multiple times;-The ability to heal defects of any size;-A low maintenance cost;-Present at least equal mechanical properties compared to traditional materials;-Present an economic advantage over traditional materials.

Major industries are using epoxy resin systems as matrices in the fabrication of multiple composite structures due to their excellent chemical, physical, and mechanical properties. Epoxies are more effective as healing agents when compared to other healing compounds in terms of cost and healing capabilities. In order for the damage recovery to be effective in large-scale composites, it must present a high restoration ability. Considering this, different studies [4,5,6] have successfully synthesized microcapsules embedding epoxy healing agents. Based on the self-healing chemistry of polymers and polymer composites, state-of-the-art approaches have categorized these materials into two classes: autonomic (or extrinsic) and non-autonomic (intrinsic). These classes are presented in Figure 1 [7,8].

Starting from the inspiring paper on liquid encapsulation by White et al. [9] for self-healing polymers, the amount of research based on autonomous (extrinsic) healing methods has highly advanced with respect to the following:-Improvement in encapsulation techniques for obtaining a higher efficiency, stronger shell walls, capsule homogeneity, capsule stability, etc. [10,11,12,13,14];-Selection of healing agents and catalyst pairs suitable for different matrices and encapsulating shell materials for the use of efficient, less expensive catalysts; use of healing agents adapted to the media of use; and use of new encapsulated chemistries, such as azide/alkene click reactions [15];-Development of healing agents that do not require cross-linker- or catalyst-like solvents, or water- and surface-reactive agents, such as silyl esters and oils [16,17,18,19,20];-Implementation of the encapsulation concept in more application-oriented research.

The occurrence of self-healing in microcapsule-based polymer and polymer composite systems has been demonstrated using fracture experiments [9,16,17,18,19,20,21,22], which have exhibited a healing efficiency as high as 75% and even reached values of up to 90%. Although a variety of alternative microencapsulation techniques are available, they are not all necessarily suited to self-healing, and the encapsulation method is often only suitable for specific types of core materials [23]. Therefore, Table 1 provides an overview of different materials that can be used as healing agents, particularly within a capsule-based system. Additionally, it can be seen that only a few healing agents have been used in fiber reinforced composite self-healing studies [24,25,26,27,28,29,30,31,32]. Of the healing systems presented in Table 1, dicyclopentadiene (DCPD) and 5-ethylidene-2-norbornene (ENB) are those which have been the most extensively investigated and are also the object of this study.

## 2. Materials and Methods

### 2.1. Manufacturing of the Poly-Urea-Formaldehyde (PUF)-Dicyclopentadiene (DCPD) Healing System

#### 2.1.1. Materials

The materials used for the synthesis of poly-urea-formaldehyde (PUF) microcapsules containing dicyclopentadiene (DCPD) as healing agent are presented in Table 2. All materials were purchased from Sigma-Aldrich (https://www.sigmaaldrich.com) and used as received.

#### 2.1.2. Synthesis of PUF-DCPD Microcapsules

Microcapsules from the same batch, synthesized by in situ polymerization, were used, as presented in [33]. In a 600 mL tall-form glass beaker, two solutions were initially prepared: a solution containing 150 mL distilled water, 7 g urea, 0.5 g resorcinol, and 0.5 g ammonium chloride, and a solution containing 100 mL co-polymer 5 wt % maleic anhydride. The two solutions were mixed until homogenization on a magnetic stirrer with a hot plate (C-MAG HS 10) at 500 rpm for 10 min. The solution pH was adjusted to 3.5 using 10 vol % sodium hydroxide and 10 vol % hydrochloric acids and the temperature was increased from room temperature to 35 °C to prevent the DCPD monomer from crystalizing when added. In order to form healing agent spherulites in the pre-polymer solution, the DCPD monomer must be in a liquid state. Considering this, in a 200 mL glass beaker, 60 g of DCPD was weighted (OHAUS PA224 analytical balance) and heated at 35 °C (Figure 2a). After the DCPD monomer became liquefied, it was poured over the previous mixture solution, under continuous stirring for approximately 5 min, to form microspheres. A formaldehyde 37% solution (0.23 mL, 18.91 g) was made and introduced after microspheres began to form. The temperature was increased to 50 °C and the solution was left under continuous stirring at 500 rpm for 2 h. During this period, the initially formed healing agent (DCPD) microspheres were wrapped in a polymeric urea-formaldehyde layer. After 20 min, whitening of the DCPD microspheres could be observed, showing that the process of coating them with urea-formaldehyde polymer was carried out under good conditions, as presented in Figure 2b. It was observed that some microcapsules remained suspended on the side of the beaker wall due to the evaporation of the water over time. To compensate for this, 200 mL distilled water was pre-heated at 50 °C and introduced in the solution. The distilled water was pre-heated to prevent crystallization of the unreacted monomer. After two hours, the solution was left to cool at room temperature while the synthesized microcapsules precipitated on the bottom. The microcapsule suspension was diluted with 200 mL distilled water and filtered under vacuum, after which the microcapsules were washed three times with 500 mL distilled water to remove traces of uncoated monomer that adhered to the surface of the microcapsules at the time of recrystallization. The suspension was then left to dry for 24 h at room temperature (21 ± 1 °C), as presented in Figure 2c.

Figure 3 presents a schematic representation of the work procedure for poly-urea-formaldehyde microcapsule synthesis with dicyclopentadiene as healing agent. During the synthesis of DCPD embedded in PUF microcapsules, at about 30 min after the addition of the monomer, a small specimen was collected to control the synthesis process (Figure 4). It was observed that microcapsules were already formed and wrapped in the polymeric layer. Subsequent to microcapsule synthetization, it was seen that, after cooling, an amount of DCPD monomer was not encapsulated, but crystallized on the surface. This can be attributed to the different stirring regime employed compared to those presented in literature reports, where blends of over 500 rpm were reported in the synthesis process. After drying, the microcapsule surface presented small lumps, but with mild stirring, the microcapsules were separated. A quantity of 17.3 g of microcapsules was obtained from this process.

Figure 5a illustrates the presence of uncoated monomer on the microcapsule surface, even after thorough cleansing. The microcapsules obtained were analyzed with the Olympus optical microscope GX and measured with a 1 cm ruler, having the unit of 100 µm (Figure 5b). SEM analysis (FEI Inspect F50) produced a better image of the uncoated monomer on the microcapsule surface, as well as the size. As can be seen in Figure 6, the microcapsule dimensions are equivalent to those reported in the literature, varying between 100 and 300 µm, with a mean value that does not exceed 250 µm.

### 2.2. Manufacturing of the Melamine-Urea-Formaldehyde (MUF)-5-ethylidene-2-norbornene (ENB) Self-Healing System

5-ethylidene-2-norbornene is an organic colorless liquid compound formed from an ethylene group bound to the norbornene molecule. The ethylene norbornene molecule consists of two unsaturated groups, one of which participates in the copolymerization process (norbornene), with the other (the ethylidene group) preceding the “vulcanization” phenomenon. The cross-linked agent (CL) is the isomer of the compound (exo-endo-isomer and exo-exo-isomer). DCPD healing agent was replaced by Lee et al. [34] with ENB, which is known for its higher reactivity. The norbornene molecule consists of a cyclohexene ring with a methylene bridge between carbon atoms 1 and 4. This monomer has a double bond that generates a greater ring resistivity and reactivity, and thus a better reactivity of ENB monomer compared to DCPD monomer. Higher healing efficiencies were reported when ENB healing agent and a 1st generation Hoveyda–Grubbs catalyst were used together. With 5 wt % of Grubbs catalyst and 10 or 20 wt % of 760 nm MUF/EMA (ethylene maleic anhydride) nanocapsules containing ENB, 97% and 123% of the fracture toughness could be recovered, respectively. Moreover, the epoxy systems used in this concept were first pre-cured during low temperature cycles, followed by a high temperature curing cycle at 170 or 180 °C [34,35,36,37,38].

#### 2.2.1. Materials

Materials used for the synthesis of melamine-urea-formaldehyde (MUF) microcapsules containing 5-ethylidene-2-norbornene (ENB) as healing agent are presented in Table 3. All materials were purchased from Sigma-Aldrich (https://www.sigmaaldrich.com) and used as received.

#### 2.2.2. Synthesis of MUF-ENB Microcapsules

Microcapsules from the same batch, synthesized by in situ polymerization, were used, as presented in [33]. The fabrication of MUF-ENB microcapsules required four solutions to be prepared. For the first solution, 0.61 g urea was introduced in 30 mL of distilled water under continuous stirring until homogenization (approximately 10 min). The second solution comprised the melamine-formaldehyde pre-polymer. In 70 mL of distilled water, 3.81 g of melamine and 6.89 g of formaldehyde (37%) were introduced and left for 25 min at 70 °C to react, and were subsequently cooled down to room temperature. The third solution was formed of 30 mL 0.5 wt % sodium lauryl sulfate (SLS) and left for 20 min at 70 °C for homogenization. Preparation of the fourth solution included 30 mL 6.3 wt % polyvinyl alcohol being left for 2 h under continuous stirring at room temperature. After the preparation of urea solution, the stirring rate was increased to 300 rpm. SLS and polyvinyl alcohol (PVA) solutions were added to the admixture and the stirring rate was increased again to 500 rpm. After homogenization (10–15 min), 30 mL of ENB was slowly poured into the admixture. As ENB is not miscible with water, under continuous stirring, it forms spherules. The stirring rate was kept constant for 10 min at room temperature, after which the temperature was increased to 86 °C within 40 min. The process of coating the microspheres obtained in the melamine-urea-formaldehyde polymer solution took place at 86 °C with continuous stirring at 500 rpm for 320 min, as shown in Figure 7. After the reaction was completed, the hot plate was turned off and the solution was left under continuous stirring until it reached room temperature, after which the reaction vessel was moved and left for 30 min until microcapsule deposition. Microcapsule emulsion was vacuum filtered and rinsed three times with 300 mL distilled water. The microcapsules were left to dry at room temperature and 21 ± 1 °C for 12 and 24 h, respectively. Figure 8 shows that after drying, there was a tendency of microcapsule agglomeration in the form of lumps, which was the same as in the case of DCPD, but these lumps were easily separated with mild stirring. The fact that the obtained microcapsules were easily separated by this mild stirring shows that the respective batch had been successfully prepared. The blue shade of the filtrate was due to the use of PVA. In Figure 9, a schematic representation of the work procedure used for melamine-urea-formaldehyde microcapsule synthesis with 5-ethylidene-2-norbornene as healing agent is presented.

During the synthesis, at about 30 min after the addition of the monomer, a sample was collected to control the synthesis process. After 12 h at room temperature, small traces of water were still visible, which eventually dried out after 24 h. The red highlighted areas in Figure 10a represent microcapsules that were broken to observe their behavior. It was observed that when the microcapsules were broken, the healing agent evaporated and the specific area tended to darken. Therefore, the process proceeded in optimal conditions and the MUF microcapsule synthesis was performed successfully. After 24 h at room temperature for drying, a quantity of 22 g of microcapsules was obtained.

In the microstructural analysis, a major difference in the size of the microcapsules was observed between the two processes. It could be easily observed that most of the obtained microcapsules were smaller than 100 μm for the MUF-ENB system. This represents a favorable outcome, as the MUF-ENB system will be easier to integrate in the matrix and composite material.

### 2.3. Fourier Transform Infrared Spectroscopy Analysis

In order to verify that the PUF-DCPD and MUF-ENB microcapsules were successfully synthesized, FT-IR analysis was performed. The synthesized microcapsules were characterized by FT-IR using a Nicolet iS50FT-IR (Nicolet, MA, USA) spectrometer equipped with a DTGS detector, providing information with a high sensitivity in the range of 400 to 4000 cm^−1^ at a resolution of 4 cm^−1^. Each spectrum was obtained by co-adding 32 scans. The analysis is presented in Figure 11, where the constituent shell and core elements can be observed. The spectra confirm that the shell material of the capsules contains urea-formaldehyde (UF) polymer. Furthermore, the FT-IR spectra of capsules entirely contain the core and shell peaks, proving the successful encapsulation of DCPD and ENB monomers in the UF/MUF shell.

### 2.4. Thermogravimetric Analysis

Thermal analyses, consisting of thermogravimetric analysis (TGA) and differential thermal analysis (DTA), were conducted simultaneously on Shimadzu DTG-TA-50H equipment at a heating rate of 10 °C. Figure 12 and Figure 13 present the TGA/DTA analysis.

The microcapsule samples were scanned in air atmosphere, at a ramp rate of 10 °C/min from ambient temperature to 500 °C, while the mass loss trace was simultaneously recorded with the equipment software. This analysis was performed in order to observe the mass loss obtained with an increasing temperature and phase changes for both PUF-DCPD and MUF-ENB systems.

The thermogravimetric analysis performed for the PUF-DCPD system indicates a total loss of mass in the range of 30–500 °C of 72.604%. The PUF-DCPD system is thermally stable up to 250 °C and thermal degradation can then be observed, which takes place in two stages up to 500 °C. It is possible that the first step, between 250 and 350 °C, is an associated process of degradation of the PUF shell, and the second stage of the interval, between 350 and 500 °C, is an associated process of degradation of the core.

The thermogravimetric analysis performed for the MUF-ENB system indicates a total mass loss in the range of 30–500 °C of 74.093%. The MUF-ENB system is thermally stable up to about 250 °C and a continuous thermal degradation process can then be observed up to 500 °C.

### 2.5. Transmission Electron Microscopy Analysis

The core-shell structure of both microcapsule types was confirmed by the TEM images presented in Figure 14. TEM analysis was performed on a Titan Themis 80–200 (Thermo Fisher (former FEI) Hillsboro, OR, USA) high-resolution transmission electron microscope, at 200 kV, with a field emission gun (FEG), courtesy of the National Research Centre for Food Safety [39]. The bright field images presented in Figure 14 present clear and spherical particles of the capsules. The size of the capsules varies from 0.2 to 2.5 µm. From the detailed images, we can see that the shell is approximately 150 nm thick.

### 2.6. Fabrication of Reference Specimens

Prior to the integration of self-healing systems in the epoxy matrix and evaluation of their mechanical properties, an investigation was conducted to find the optimal matrix curing cycle. Different curing cycles were investigated for the epoxy system Resoltech 1050 and Hardener 1055S, in order to achieve the best mechanical properties; firstly, according to the technical data parameters (16 h at 60 °C) and secondly, by increasing the temperature to 80 °C and reducing the curing time to 2 h. Curing was done in a POL-EKO 240 SLN oven.

Neat epoxy reference specimens were fabricated according to SR EN ISO 14125:2003 (Class IV) and were compared with specimens containing PUF-DPCD and MUF-ENB self-healing systems. Any specimen with a measured thickness (h) exceeding ±2% tolerance was eliminated. Additionally, the width (b) and thickness (h) were measured with a 1% accuracy. The length (L) was adjusted by exactly 1% of the calculated value to match the distance between the support grips and average specimen thickness (L/h), according to the standard.

### 2.7. Finite Element Method Analysis

Detailed structural analysis regarding the mechanical behavior of composite materials has gained a lot of interest in recent years, especially with their increasing usage in almost all major industries. This type of analysis has become much faster and cheaper to perform compared to experimental methods that are expensive and time-consuming. The numerical approach by means of the finite element method (FEM) offers the advantage of a faster, cheaper, and more detailed analysis for an infinite number of material configurations, especially composite materials [40].

Therefore, a numerical analysis approach was taken into consideration for this paper, in order to obtain a better understanding of the microcapsule behavior when embedded in an epoxy matrix. This will further help in the investigation and analysis of the mechanical behavior of self-healing composites when subjected to different loads and will be the subject of another paper.

According to elasticity theory, the presence of a microcapsule within a given material represents an inclusion and after its rupture, the broken microcapsule creates a stress concentration phenomenon. For the analysis of this stress concentration, a cube-cell with a side of 0.2 mm and a 20 µm microcapsule embedded was designed, as shown in Figure 15a. The material chosen for the cube-cell was the same Resoltech epoxy resin used for the microcapsule integration in Section 3. Tetrahedron elements with 10 nods were used for the analysis, having a 2nd degree order of interpolation, which increases the analysis accuracy. This model is presented in Figure 15b.

The cube-cell was tensile stressed until it broke at 125 MPa. To model the healing agent, an elastic material with a very low Young’s Modulus was chosen and a 0.495 Poisson coefficient was employed to represent the liquid incompressibility. The Von Mises stress distribution is presented in Figure 16.

For the filled microcapsule, the maximum stress was found at 150 MPa, with a stress concentration factor (K_c_) of 1.2. After microcapsule rupture, the maximum stress was found at 242 MPa, with a stress concentration factor of 1.94. The analysis performed for the 20 µm microcapsule is presented in Figure 17 The maximum stress caused by the spherical inclusion can be found at 624 MPa, corresponding to a stress concentration factor of 1.49.

Another aspect of this numerical analysis was the effect on the matrix mechanical loads for the relative position of two microcapsules, while embedded within the matrix. The first case was the arrangement of two microcapsules at a distance of 0.085 mm at an angle of 45 degrees. In this case, the maximum stress was 795 MPa and the stress concentration factor was 1.87, as illustrated in Figure 18a. The second case was characterized by the disposition of the two microcapsules in the median plane, disposed in the normal direction of the tensile force. The stress distribution is shown in Figure 18b. In this case, the maximum stress was 818 MPa and the stress concentration was 1.92. The third case was characterized by the microcapsule position overlapping on the plane perpendicular to the direction of the tensile force. It was found that the maximum stress was 786 MPa and the stress concentration factor was 1.85, as presented in Figure 18c.

The presence of two microcapsules caused the concentration coefficient to increase by 29% in the most unfavorable case, corresponding to a decrease of the breaking force of 23%. Moreover, increasing the spherical inclusion diameter by two times induced an increase of the concentration factor from 1.2 to 1.49 (25%). This represents a 20% decrease in the breaking force with respect to classical failure theories.

The maximum tensile strength of the cube-cell without the spherical inclusion is defined by Equations (1)–(4):(1)$$\sigma_{r}=\frac{F_{r}}{A}.$$

In the presence of the inclusion, the tensile strength is
(2)$$\sigma_{r}^{c}=k\frac{F_{r}^{c}}{A}=\frac{F_{r}}{A}.$$

The expression of the breaking force in the case of an inclusion is as follows:(3)$$F_{r}^{c}=\frac{F_{r}}{k}=\frac{1}{1.25}F_{r}=0.8F_{r}.$$

According to the Finite Element Analysis (FEA), the breaking force calculated with classical theories does not cause the rupture tensile to occur in the entire section, but only in certain regions of the interface between the spherical inclusion and matrix, with theories of fracture mechanics being applicable. Increasing the value of the tensile above the ultimate strength will cause a micro-crack to appear at the interface between the spherical inclusion and the matrix in the perpendicular plane in the direction of the tensile force, in the case of tensile breaking tests. Compared with the concentration factor of a void (3 in the case of a sphere), the existence of the liquid reduces the concentration factor by half (1.5).

Using Reuss’ rule for the mechanical properties of a material consisting of two constituents, we have the following relations:(4)$$E=f_{1}E_{1}^{}+f_{2}E_{2}^{}\approx f_{1}E_{1}^{},$$
where *E* is the elasticity modulus of the mixture, *E_i_* is the elasticity modulus of the i component, *f_i_* is the volumetric fraction of the i component, and the modulus of elasticity of the liquid is much smaller than that of the matrix (it is considered equal to 0).

## 3. Results

### 3.1. Integration of PUF-DCPD in Epoxy Specimens

Following the curing cycle of neat epoxy specimens, six integration methods were performed to establish an optimal integration process, as presented in Table 4.

#### 3.1.1. Method 1 and Method 2

Following the first integration method and taking into consideration the viscosity of RESOLTECH 1050 resin, which is 1043 mPa.s at 23 °C, proper integration of the microcapsules was difficult to obtain. Additionally, due to the large size of PUF-DCPD microcapsules and the resulting high volumetric fraction, the integration of 15 wt % microcapsules into the epoxy resin was difficult to achieve. Because of this high volumetric fraction, the mixture could no longer be injected into the silicon molds, causing very large pores or a lack of material uniformity, which are defects that can be seen in Figure 19.

For the second method, the matrix viscosity was reduced by heating the epoxy resin to 40 °C while maintaining the stirring rate at 300 rpm for homogenization. By doing so, the microcapsule integration time was halved compared to method 1 and the mixture could be injected into the silicon molds to manufacture specimens at a curing cycle of 80 °C for 2 h. It was found that most microcapsules have a tendency to rise at the surface of the specimens, as seen in Figure 19b. The specimens fabricated using method 1 and method 2 presented a high elasticity and therefore, a loss in the matrix mechanical properties, as a result of the microcapsules breaking during homogenization at 300 rpm. When the microcapsules break, the cross-linked agent, DCPD, is released into the epoxy resin and because the monomer does not cross-link without a hardener, it increases the host’s elasticity.

#### 3.1.2. Method 3 and Method 4

For methods 3 and 4, the homogenization rate was decreased to 200 and 100 rpm, respectively, to reduce the number of broken microcapsules and thus the elasticity given to the specimens. A slight increase of their aspect and mechanical properties was obtained, but was insufficient for three-point bending tests, as the specimens could not reach the minimum 5N preload stated in the standardized method. Considering the epoxy matrix curing cycle at 60 °C for 16 h, a pre-polymerization step for the epoxy matrix when integrating the microcapsules was proposed, in order to avoid the agglomeration of microcapsules on the surface.

#### 3.1.3. Method 5 and Method 6

In method 5, the epoxy matrix was preheated at 60 °C, the hardener was added, and the stirring rate was set to 100 rpm, after which 15 wt % microcapsules were introduced into the mixture. After homogenization, the epoxy resin-containing microcapsules were injected into silicon molds and cured at 80 °C for 2 h.

Method 6 consisted of pre-heating the epoxy resin at 60 °C. To further reduce its viscosity, 10 wt % acetone was added before introducing the hardener, followed by homogenization at 100 rpm and the addition of 15 wt % microcapsules. The temperature was raised to 80 °C and the stirring rate was kept at 100 rpm until acetone evaporation. Then, microcapsules were introduced in silicon the mold and cured at 80 °C for 2 h. This method was found to be the most efficient for the integration process. The same procedures were applied for MUF-ENB microcapsules, where method 5 was considered the most efficient. The fabricated specimens are presented in Figure 20.

### 3.2. Thermal Stability Study for PUF-DCPD and MUF-ENB Self-Healing Systems

In order to determine the thermal stability of the synthesized microcapsules, a series of tests were performed with respect to the curing cycles of the final composite material, in which the self-healing systems were integrated according to method 6 for PUF-DCPD microcapsules and method 5 for MUF-ENB microcapsules. Therefore, thermal stability tests were performed at 60, 80, and 120 °C for an interval of 1, 2, 3, and 4 h.

#### Testing Procedure

The oven was preheated to the testing temperature (60, 80, and 120 °C). Four empty beakers were weighted, along with four 0.20 g samples, for each beaker. Each beaker containing the sample was exposed to different periods of time, namely, beaker 1—one-hour exposure, beaker 2—two-hour exposure, beaker 3—three-hour exposure, and beaker 4—four-hour exposure. Each of the four beakers were left in a desiccator after they were removed from the oven to cool down, after which they were weighted to determine the weight loss obtained by evaporation of the healing system. The difference between the weight before temperature exposure and after exposure represents the total weight loss percentage and is presented in Figure 21 and Figure 22.

All samples were analysed under an optical microscope after being exposed to temperature, in order to determine the thermal stability, to ensure that the microcapsules retained their structure. It was observed that the majority of the PUF-DCPD and MUF-ENB microcapsules were not damaged, with the mass loss following the exposure to temperature taking place by evaporation of the repair agents dicyclopentadiene and ethylidene-norbonene, respectively, through the walls of the microcapsules. Although, in the case of exposure to the temperature of 120 °C, which is the temperature related to the polymerization cycle of the final composite material, the mass loss is substantial, this does not greatly influence the efficiency of the integrated healing systems, since the host matrix at 120 °C is already cured, having passed through a prepolymerization cycle at 60 and 80 °C, respectively.

### 3.3. Three-Point Bending Test Campaign

#### 3.3.1. Testing Procedure

The testing method for both reference specimens containing neat epoxy resin and specimens containing healing systems followed ISO 14125:2003 (Class IV). All fabricated specimens correspond to Class IV materials, having dimensions of 100 × 15 × 2 mm. The loading speed was calculated using Equation (1), and a 2 mm/min speed was determined. The flexural stress is given by Equation (6) and the flexural modulus is given by Equation (9), calculated by measuring the deflection points (s′ and s″), in Equations (7) and (4).
(5)$$v=\frac{\varepsilon^{\prime }L^{2}}{6h},$$
where *L* is the support span (mm), *h* is the specimen thickness (mm), and *ε′* is the strain rate of 0.01/min or 1%/min.
(6)$$\sigma_{f}=\frac{3FL}{2bh^{2}},$$
where $\sigma_{f}$ is the flexural stress (MPa), *F* is the applied force (N), *L* is the support span (mm), *h* is the specimen thickness (mm), and *b* is the width of the specimen (mm)
(7)$$s^{\prime }=\frac{\varepsilon_{f}^{\prime }L^{2}}{6h},$$
(8)$$s^{\prime \prime }=\frac{\varepsilon_{f}^{\prime \prime }L^{2}}{6h},\;$$
(9)$$E_{f}=\frac{L^{3}}{4bh^{3}}\left(\frac{∆F}{∆s}\right),$$
where, *s′* and *s″* represent the deflection in the middle of the specimen (mm), $\varepsilon_{f}^{\prime }$ is the flexural stress for deflection *s′* and its corresponding value is 0.0005, and $\varepsilon_{f}^{\prime \prime }$ is the flexural stress for deflection *s″* and its corresponding value is 0.0025.

#### 3.3.2. Testing of Neat Resin Specimens

Three-point bending tests were performed with Instron 3360 Series Universal Testing Systems. The load was applied to the specimens at mid-span until rupture. Table 5 and Table 6 present the flexural stress at break values for specimens cured at 60 °C for 16 h and at 80 °C for 2 h, respectively, and the corresponding stress–strain curves are presented in Figure 23 and Figure 24.

#### 3.3.3. Testing of PUF-DCPD Specimens Fabricated by Method 5

Specimens obtained by method 5 showed an easier integration of microcapsules and maintained their dispersion after the curing cycle. Although the amount of broken microcapsules decreased during homogenization, the specimens displayed large differences during the three-point bending tests, as shown in Table 7 and Figure 25.

#### 3.3.4. Testing of PUF-DCPD Specimens Fabricated by Method 6

The specimens obtained through method 6 showed a good dispersion of the microcapsules in their structure and mechanical properties comparable to those of neat resin specimens with the same curing cycle (Table 8 and Figure 26).

#### 3.3.5. Testing of MUF-ENB Specimens Fabricated by Method 5

Table 9 presents the results of flexural tests performed for each MUF-ENB specimen. Figure 27 shows the stress–strain curves for tested specimens. The support span of each specimen was measured at 80 mm.

#### 3.3.6. Self-Healing Evaluation

Materials used in the specimen manufacturing process were M9.6GF/37%/300H8/G prepreg (glass fiber reinforced), Resoltech 1050/1058 epoxy resin (compatible with prepreg matrix), PUF-DCPD, MUF-ENB, and first generation Grubbs catalyst. Composite specimens containing a 15 vol % PUF-DCPD system and 10 vol % MUF-ENB system, respectively, were fabricated with respect to method 6 and in accordance with ISO 14125:2003 (Class IV).

For the specimens containing the PUF-DCPD system, three-point bending tests showed a 4% average decrease in flexural strength, while for the specimens containing the MUF-ENB system, an average of a 10% decrease was observed when compared to the neat specimens. These values are acceptable, as these microcapsules are considered an induced defect in the composite structure. Moreover, specimens containing the PUF-DCPD system showed an average displacement of 13%, whilst for MUF-ENB specimens, an increase of 90% was observed when compared to neat specimens. This may be due to the larger amount of MUF-ENB microcapsules used for the specimen fabrication, as their dimensions are considerably smaller than those of PUF-DCPD microcapsules. Specimens were tested until the load was constant, which is the starting point of material degradation. Tested specimens were introduced in an oven (ambient atmosphere) at 40 °C, left for 48 h, and retested in the same conditions. After 48 h, specimens containing PUF-DCPD showed an average flexural strength of 81% compared to the reference specimens, and those containing MUF-ENB exhibited a value of 77%. Retested specimens presented an average displacement of 23% for PUF-DCPD specimens and 109% for MUF-ENB, respectively, until full specimen rupture.

Due to the large number of tested specimens and in order to obtain a better understanding of the results, the load–displacement curves are presented as two separate charts, as seen in Figure 28 and Figure 29.

## 4. Discussion

Because the stirring rate is the main factor of microcapsule formation, the stirring rate was raised to 500 rpm to overcome the traces of non-encapsulated DCPD, thus halving the unreacted monomer. Adding acetone to the process of washing the microcapsules helps remove small traces of unreacted monomer and dries the capsules faster. The same principles were applied for MUF-ENB healing systems.

After an evaluation of the six proposed methods for the integration of healing systems in the epoxy matrix, it was concluded that method 6 showed the optimal results for PUF-DCPD microcapsules and method 5 was found to be optimal for MUF-ENB microcapsules.

Three-point bending tests were performed to evaluate the optimal integration methods. When compared to neat epoxy specimens, PUF-DCPD specimens showed a 20% average decrease in flexural strength and MUF-ENB specimens displayed a 40% decrease. Due to the smaller dimensions of the MUF-ENB microcapsules obtained, the defects induced in the matrix through the integration of the system should be smaller, being directly proportional to the mechanical strength of the neat material. However, the fact that the results obtained during the three-point bending test of the specimens with the MUF-ENB self-healing system are smaller than those of the PUF-DCPD specimens shows that the integration method can be further optimized.

The results of the thermal stability tests demonstrate that the MUF-ENB microcapsules are two times more stable than the PUF-DCPD microcapsules. Therefore, for the thermal stability at 60 °C, the average weight loss for PUF-DCPD microcapsules is 20.49%, compared to only 8.39% in the case of MUF-ENB microcapsules. When exposed to the temperature of 80 °C, the PUF-DCPD microcapsules had an average weight loss percentage of 13.93%, compared to only 9.63% in the case of MUF-ENB microcapsules, and in the case of thermal stability testing at 120 °C, there was a 57.49% drop in mass for UPF-DCPD microcapsules, compared to only 19.23% for MUF-ENB microcapsules.

To evaluate the healing efficiency of the proposed integration methods, PUF-DCPD and MUF-ENB microcapsules were integrated along with the Grubbs’ catalyst in glass fiber reinforced polymer (GFRP) specimens and subjected to three-point bending tests. An initial test was conducted to evaluate the difference in flexural strength for the specimens containing healing systems when compared to the neat epoxy specimens. The PUF-DCPD and MUF-ENB specimens were retested after conditioning at 40 °C for 48 h, from the starting point of material degradation, at which the specimens were initially tested. The results showed a healing efficiency of 81% for the PUF-DCPD specimens and 77% for MUF-ENB specimens, when compared to the reference samples.

## 5. Conclusions

Within this work, the synthesis of self-repair system components was conducted and methods for their integration in the host polymeric (epoxy) matrix were developed.

Two different microcapsule-based regenerative systems whose shells were made of poly-urea-formaldehyde with embedded dicyclopentadiene monomer and melamine-urea-formaldehyde with embedded ethylidene-norbonene, were studied and compared in this work. Different integration methods were performed to identify an optimal integration process of both PUF-DCPD and MUF-ENB systems. Following identification of the optimal method, flexural tests were conducted to validate the integration methods. FT-IR analysis was conducted to validate the presence of constitutive elements in the obtained PUF-DCPD and MUF-ENB microcapsules. Additionally, thermogravimetric analysis showed thermal degradation of 73.6% for the PUF-DCPD system and 74% for MUF-ENB. Microstructural analysis was performed to observe the synthesized microcapsules and to confirm the core-shell structure.

The three-point bending test results for the evaluation of self-healing efficiency confirmed the healing capacity of the proposed approach, where at least a 77% healing efficiency was obtained. Moreover, the displacement values for the retested specimens strengthened the healing ability of the optimized integration method.

## Figures and Tables

**Figure 1 polymers-12-01052-f001:**
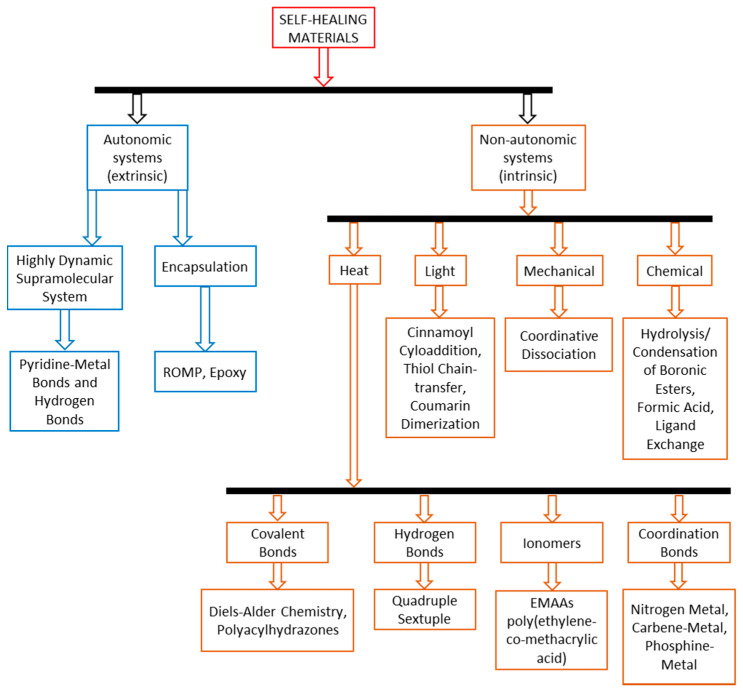
Polymer material classification with respect to their healing chemistry: autonomic self-healing systems and non-autonomic systems (adapted from [7]).

**Figure 2 polymers-12-01052-f002:**
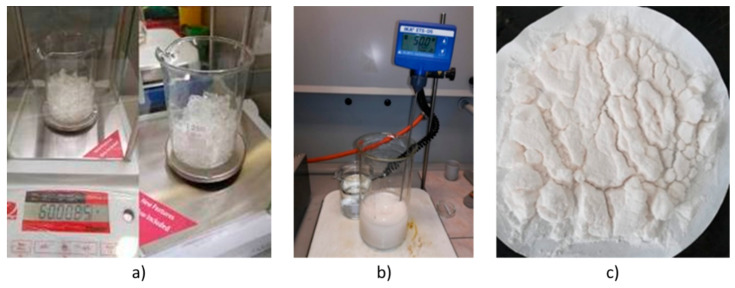
Synthetization process of PUF-DCPD microcapsules showing the (**a**) weighting of DCPD monomer at room temperature before phase transformation at 35 °C, (**b**) microcapsule formation at 50 °C, and (**c**) microcapsule suspension left to dry at room temperature for 24 h.

**Figure 3 polymers-12-01052-f003:**
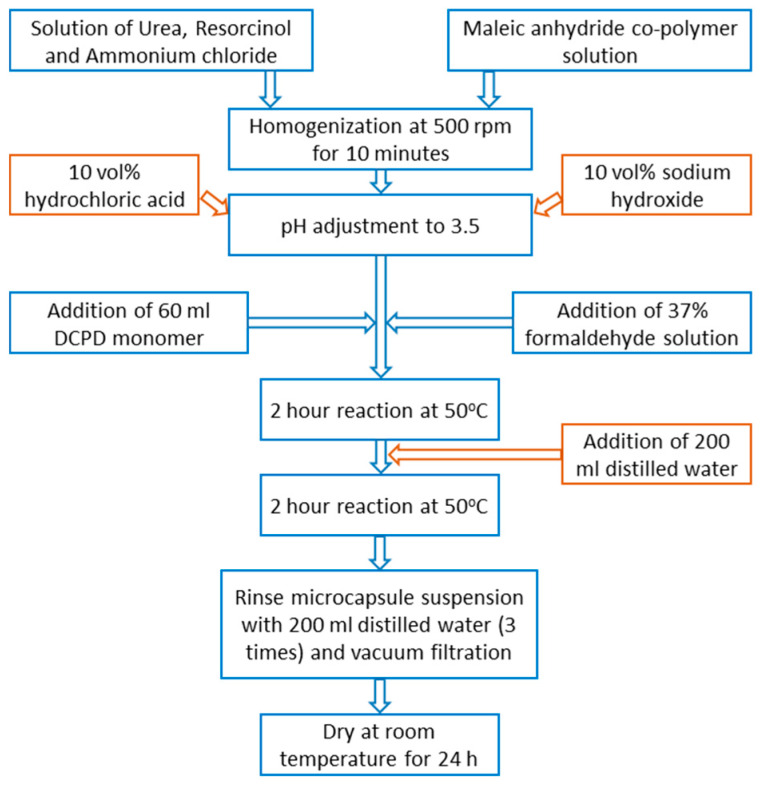
Process of PUF-DCPD microcapsule synthesis.

**Figure 4 polymers-12-01052-f004:**
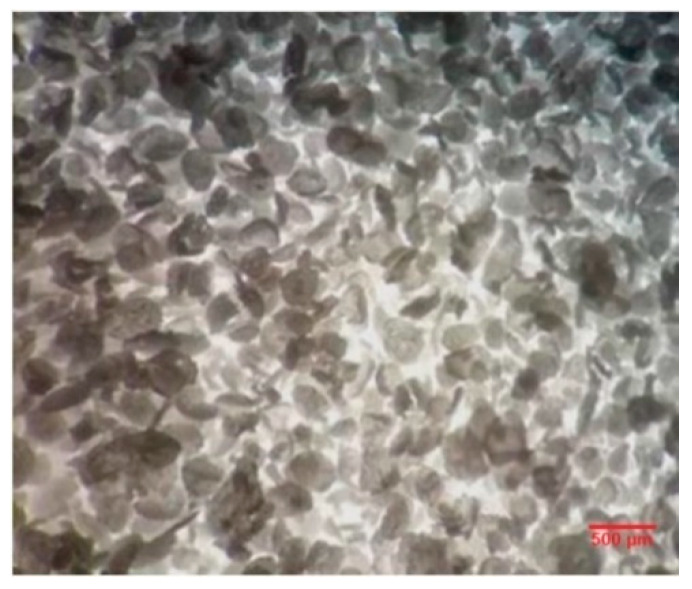
Sample taken during microcapsule synthesis process (optical image, 25× zoom).

**Figure 5 polymers-12-01052-f005:**
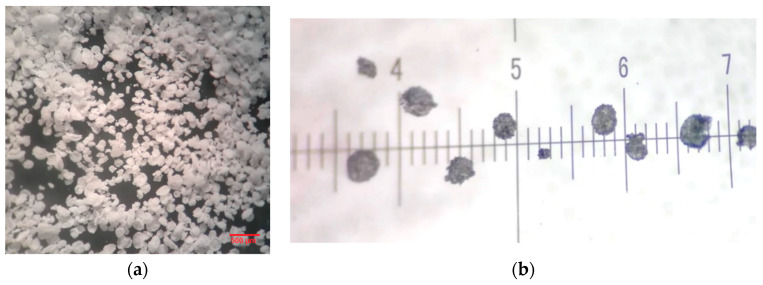
Optical images illustrating (**a**) microcapsules after 24 h drying at room temperature—25× zoom and (**b**) microcapsule dimensions—45× zoom (one division represents 100 µm).

**Figure 6 polymers-12-01052-f006:**
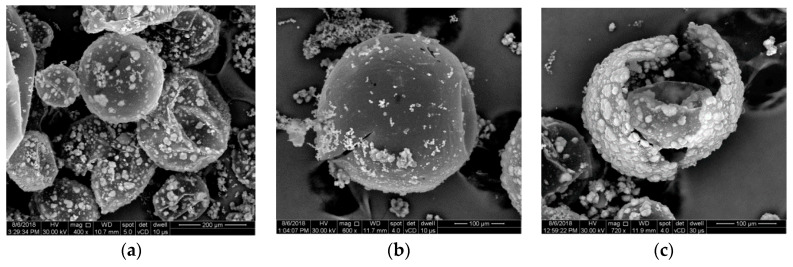
SEM images of (**a**) microcapsule agglomeration, (**b**) a 250 µm microcapsule, and (**c**) a broken microcapsule.

**Figure 7 polymers-12-01052-f007:**
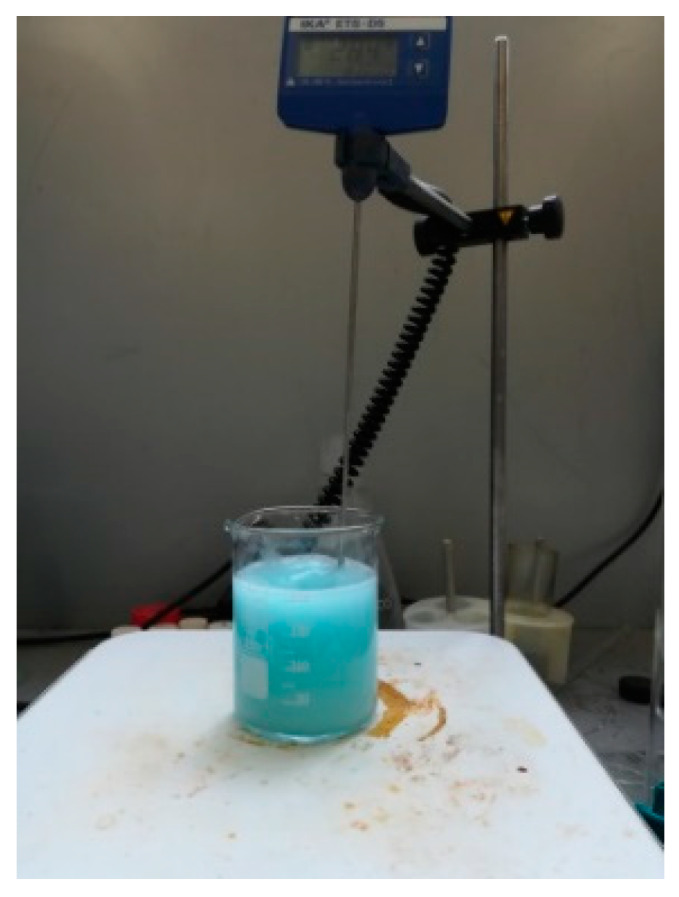
The reaction mixture employed for obtaining MUF microcapsules with embedded ENB healing agent.

**Figure 8 polymers-12-01052-f008:**
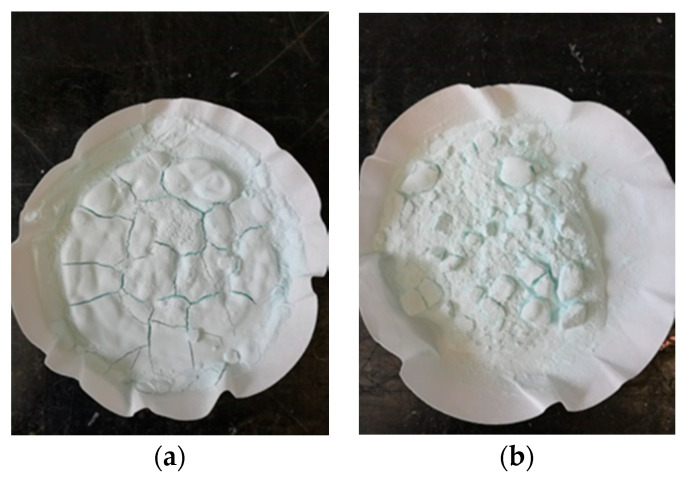
Microcapsules after a (**a**) 12 h and (**b**) 24 h drying period.

**Figure 9 polymers-12-01052-f009:**
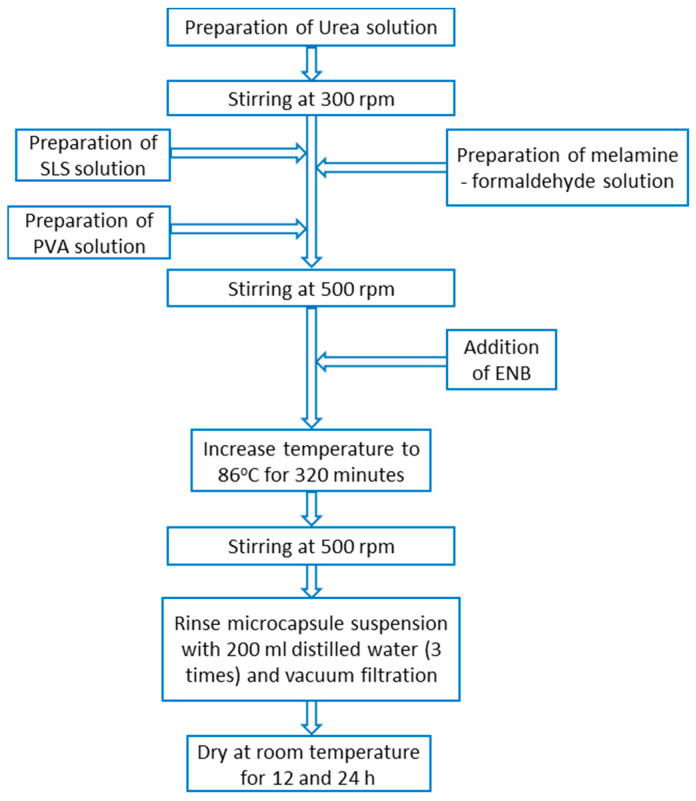
Schematic illustration of melamine-urea-formaldehyde microcapsule synthesis with embedded 5-ethylidene-2-norbornene as repair agent.

**Figure 10 polymers-12-01052-f010:**
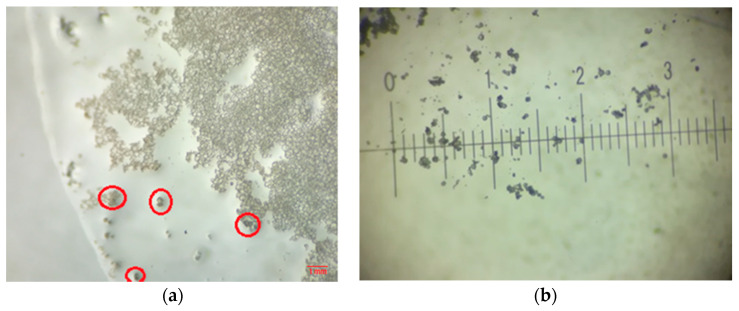
Optical image illustrating a (**a**) sample taken 30 min after the addition of ENB and a (**b**) sample after 12 h drying at room temperature (one division represents 100 µm).

**Figure 11 polymers-12-01052-f011:**
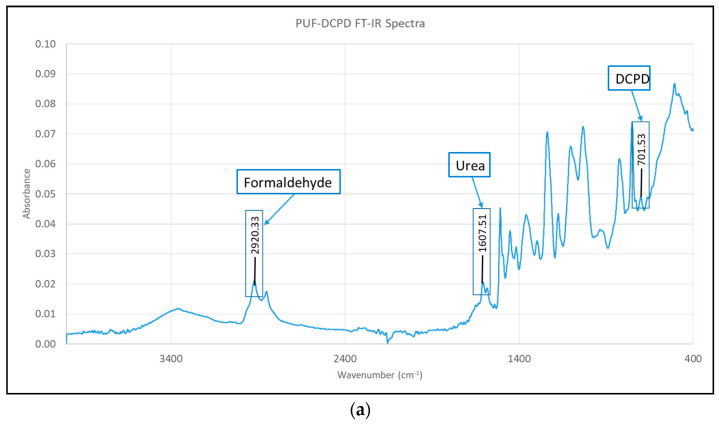
FT-IR spectra of the main (**a**) PUF-DCPD and (**b**) MUF-ENB microcapsule constituents.

**Figure 12 polymers-12-01052-f012:**
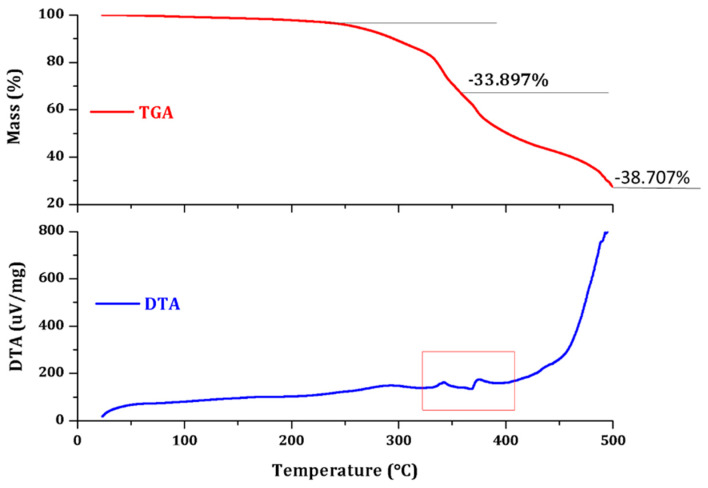
Thermogravimetric analysis (TGA)/differential thermal analysis (DTA) analysis of the PUF-DCPD system.

**Figure 13 polymers-12-01052-f013:**
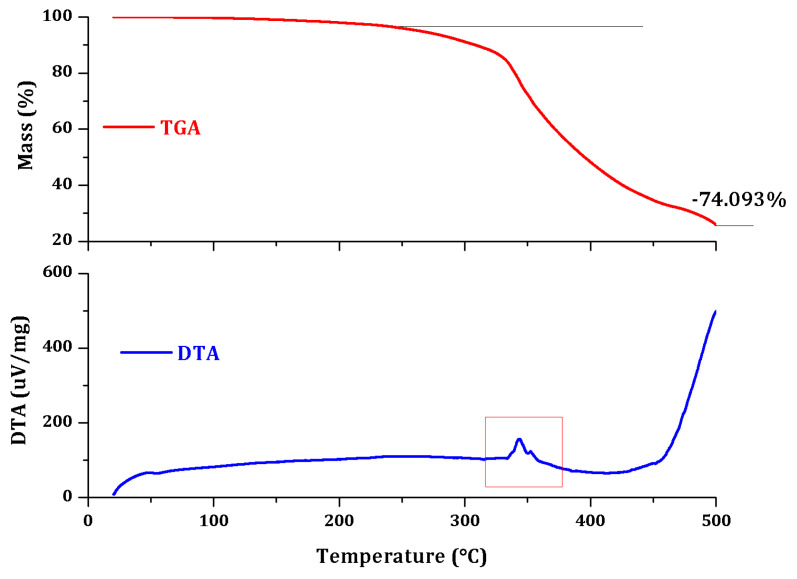
TGA/DTA analysis of the MUF-ENB system.

**Figure 14 polymers-12-01052-f014:**
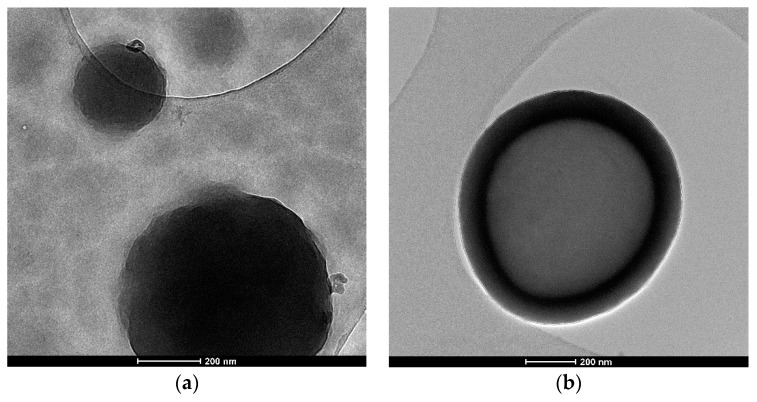
TEM images presenting the core-shell structure of synthetized (**a**) PUF-DCPD and (**b**) MUF-ENB microcapsules.

**Figure 15 polymers-12-01052-f015:**
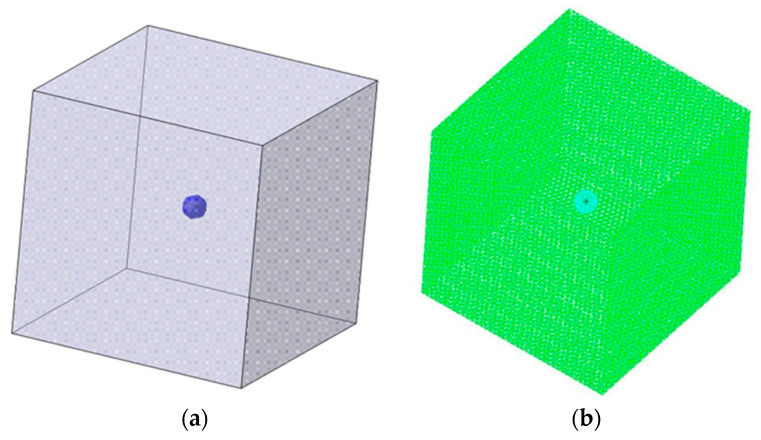
Illustration of the (**a**) cube-cell element and (**b**) finite element model with an embedded microcapsule.

**Figure 16 polymers-12-01052-f016:**
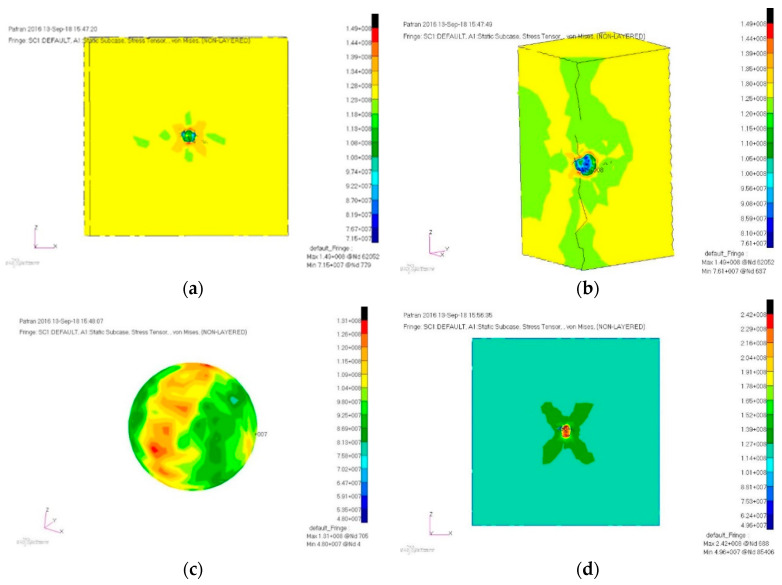
Von Mises illustration of the (**a**) cube-cell cross-section view, (**b**) cube-cell quarter view, (**c**) filled microcapsule, and (**d**) base cube-cell with an empty microcapsule.

**Figure 17 polymers-12-01052-f017:**
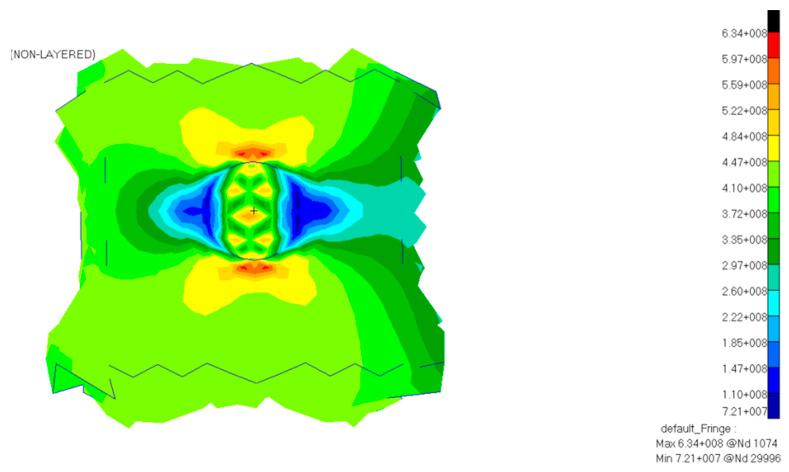
Von Mises distribution on the 20 µm microcapsule.

**Figure 18 polymers-12-01052-f018:**
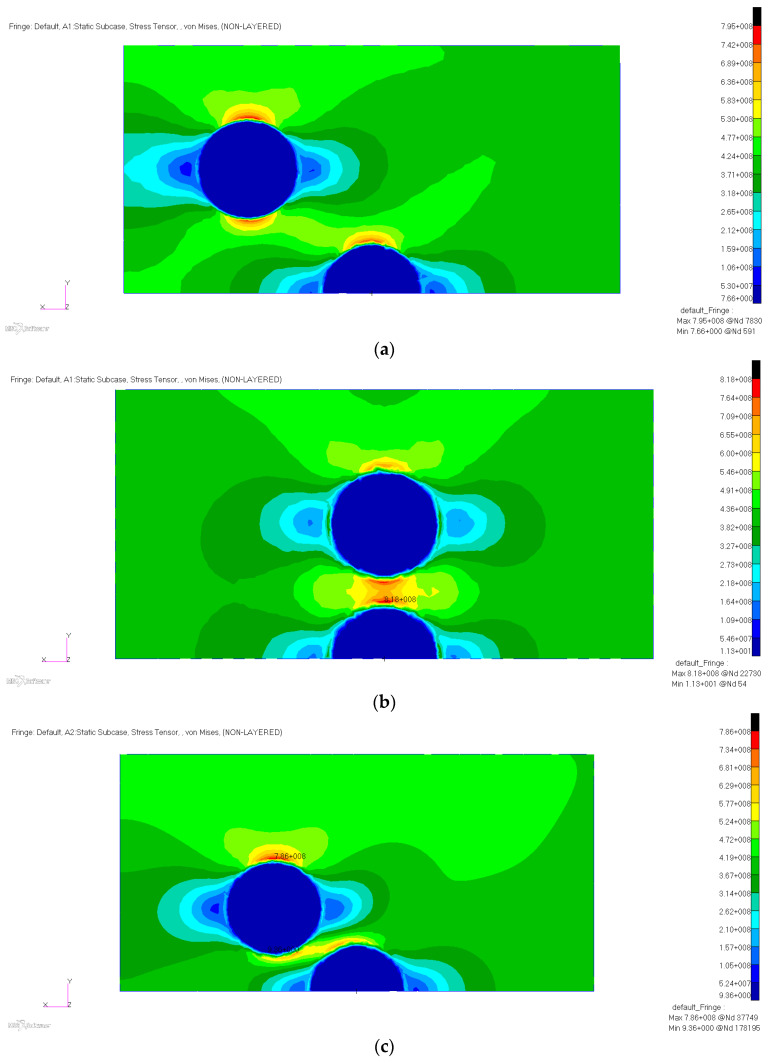
Von Mises stress distribution with respect to spherical inclusions (**a**) placed at 45° at 0.085 mm, (**b**) placed in the median plane at 0.05 mm, and (**c**) in an overlapping position in the perpendicular plane at 0.05 mm.

**Figure 19 polymers-12-01052-f019:**
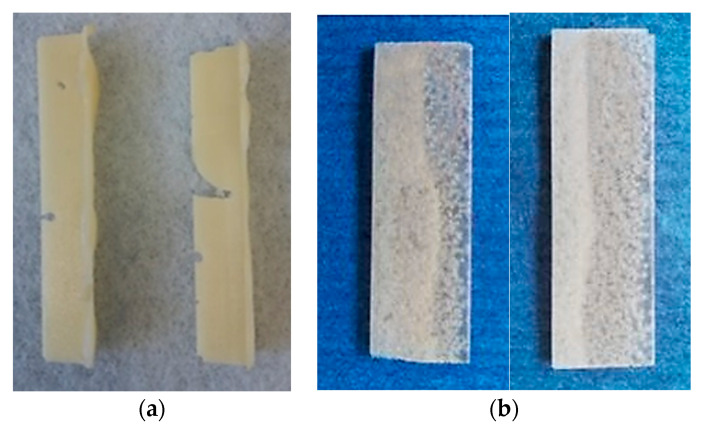
Specimens fabricated using (**a**) method 1, showing defects due to the high volumetric fraction of epoxy matrix and 250 µm PUF-DCPD microcapsules, and (**b**) method 2, presenting microcapsule agglomeration at the surface.

**Figure 20 polymers-12-01052-f020:**
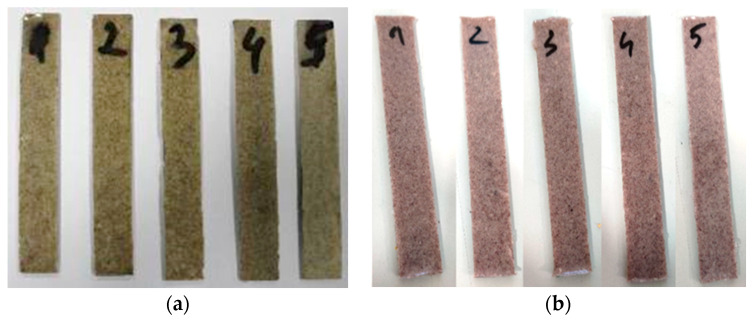
Specimens containing (**a**) MUF-DCPD and (**b**) PUF-ENB systems and Grubbs catalyst fabricated by method 6.

**Figure 21 polymers-12-01052-f021:**
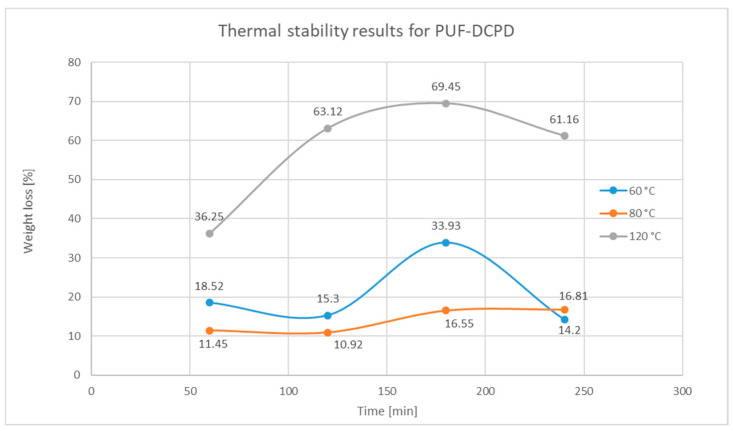
Weight loss variation over time for PUF-DCPD after thermal exposure at 60, 80, and 120 °C.

**Figure 22 polymers-12-01052-f022:**
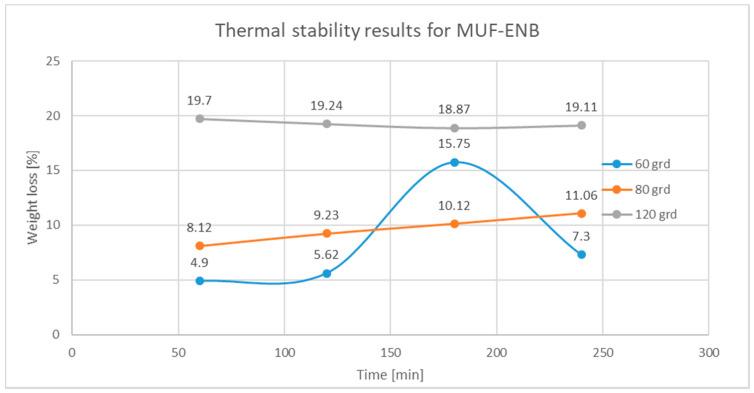
Weight loss variation over time for MUF-ENB after thermal exposure at 60, 80, and 120 °C.

**Figure 23 polymers-12-01052-f023:**
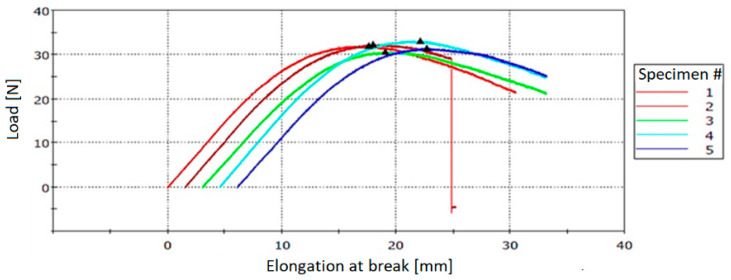
Stress–strain curves for specimens cured at 60 °C for 16 h.

**Figure 24 polymers-12-01052-f024:**
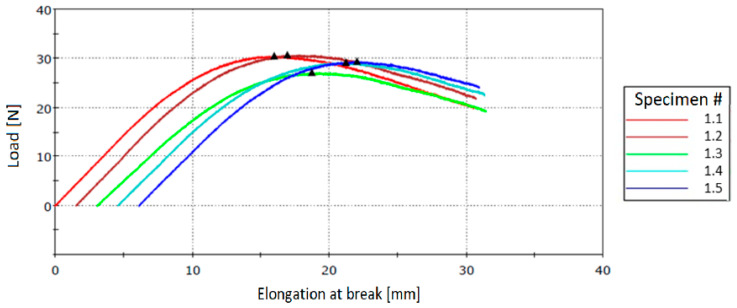
Stress–strain curves for specimens cured at 80 °C for 2 h.

**Figure 25 polymers-12-01052-f025:**
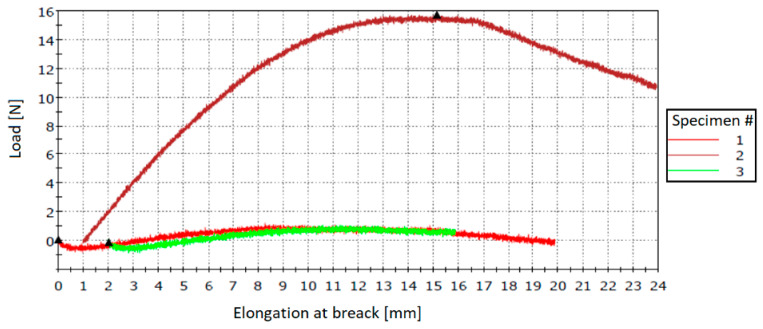
Stress–strain curves for PUF-DCPD specimens fabricated with method 5.

**Figure 26 polymers-12-01052-f026:**
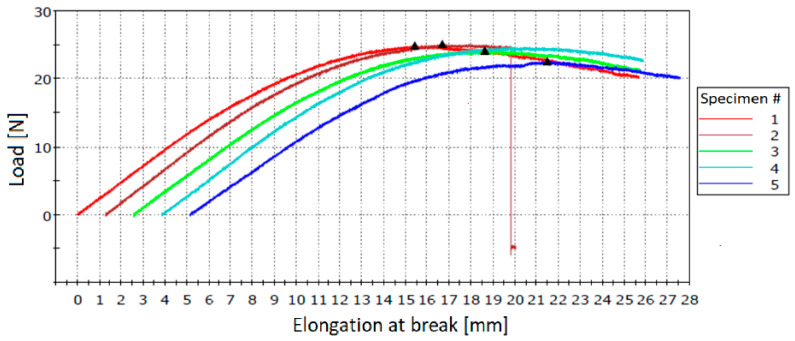
Stress–strain curves for PUF-DCPD specimens fabricated with method 6.

**Figure 27 polymers-12-01052-f027:**
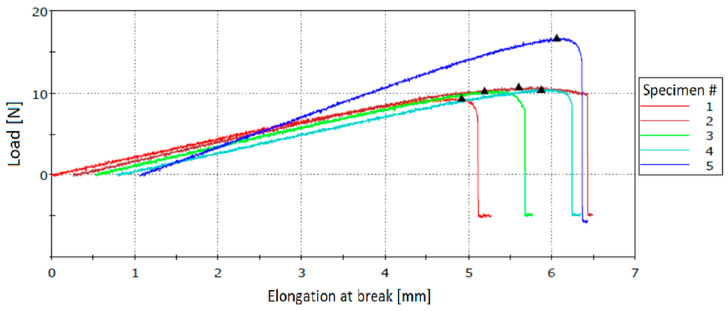
Stress–strain curves for MUF-ENB specimens fabricated according to method 6.

**Figure 28 polymers-12-01052-f028:**
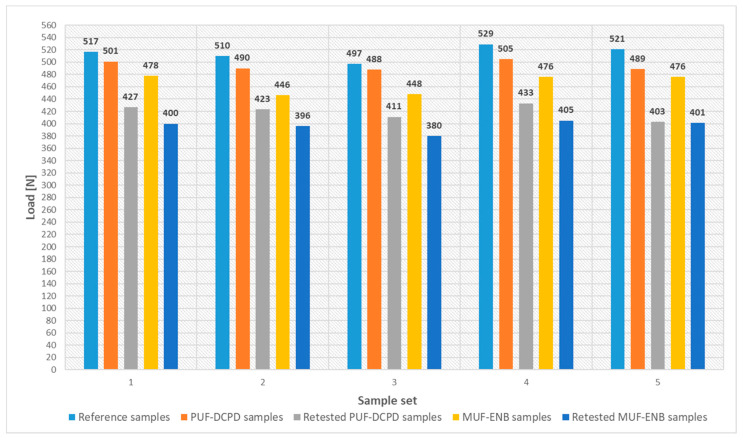
Load values for the three-point bending tested specimens.

**Figure 29 polymers-12-01052-f029:**
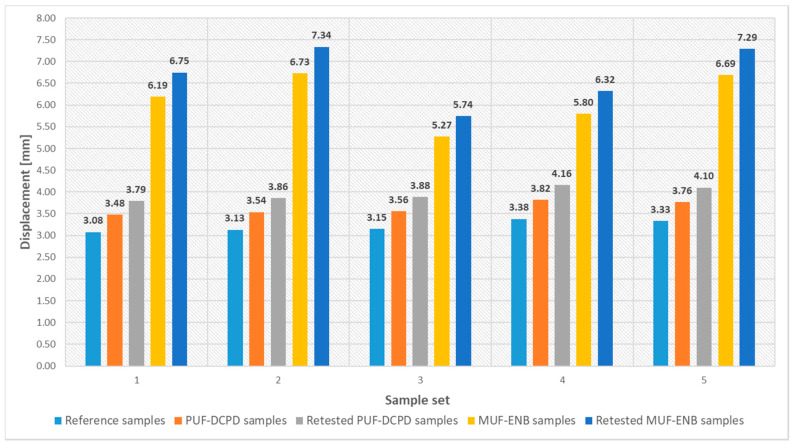
Displacement values for the three-point bending tested specimens.

**Table 1 polymers-12-01052-t001:** Overview of materials that can be used as healing agents and their respective delivery system.

Healing Agent	Delivery System			Catalyst Required	Used in FRP Composites
	Microcapsule	HGF	Microvascular		
DCPD/ENB	X (UF)		X	X	X
Siloxanes	X (UF/PU)			X	
Epoxy	X (UF)			X	X
Amine-Epoxy	X (UF)	X	X		X
Thiol-epoxy	X (MF)			X	X
Thiol-ene	X (UF)			X	
Thiol-isocyanate	X (MF/PU)				
Azide-alkyne	X (UF)			X	
Methacrylates		X		X	
GMA	X (MF)				
Melaimides	X (UF)				X
Isocyanates	X (PU)	X			
Cyanoacrylates	X	X			X
Vinyl ester	X			X	X
Unsaturated polyester	X			X	X

**Table 2 polymers-12-01052-t002:** Chemical substances used in the production of poly-urea-formaldehyde (PUF) microcapsules with embedded dicyclopentadiene (DCPD).

Material	Molecular Formula	Physical Properties	Role
Urea	CH_4_N_2_O	Crystalline, white powder. Melting point at 133–135 °C	Formation of the capsule shell in the aqueous state
Resorcinol (1,3- benzenediol)	C_6_H_4_(OH)_2_	White crystals. Melting point at 113 °C	Blending resin with formaldehyde
Formaldehyde	CH_2_O	Colorless aqueous solution	Formation of capsule shell
Dicyclopentadiene (DCPD)	C_10_H_12_	Solid state (gel-like state at room temperature). Melting point at 32.5 °C	Monomer-capsule core
Maleic anhydride	C_2_H_2_(CO)_2_O	Solid state. White powder	Emulsifier
Ammonium chloride	NH_4_Cl	Solid state. White powder	Formaldehyde hardener
Sodium hydroxide	NaOH	Solid state. White powder	Rising solution pH
Hydrochloric acid	HCl	Aqueous solution with strong odor	Lowering solution pH

**Table 3 polymers-12-01052-t003:** Chemical substances used in the production of melamine-urea-formaldehyde (MUF) microcapsules with embedded 5-ethylidene-2-norbornene (ENB).

Material	Molecular Formula	Physical Properties	Role
Melamine	C_3_H_6_N_6_	White powder. Melting point at 345 °C	Formation of capsule shell
Urea	CH_4_N_2_O	Crystalline white powder. Melting point at 133–135 °C	Formation of the capsule shell in the aqueous state
Formaldehyde	CH_2_O	Colorless aqueous solution	Formation of capsule shell
Sodium lauryl sulfate (SLS)	CH_3_(CH_2_)_10_CH_2_ (OCH_2_CH_2_)_n_OSO_3_Na	Powder soluble in water	Emulsifier—oil solidification
Polyvinyl alcohol (PVA)	(C_2_H_4_O)x	Melting point at 200 °C	Separation film

**Table 4 polymers-12-01052-t004:** Methods of microcapsule integration in the epoxy matrix.

Method No.	Method for Optimization
1	Homogenization at 300 rpm
2	Heating at 40 °C and homogenization at 300 rpm
3	Heating at 40 °C and homogenization at 200 rpm
4	Heating at 40 °C and homogenization at 100 rpm
5	Preheating at 60 °C, homogenization at 100 rpm, and heating at 80 °C
6	10 vol % acetone dilution, heating at 60 °C, and homogenization at 100 rpm, and then acetone evaporation at 80 °C

**Table 5 polymers-12-01052-t005:** Flexural tests for specimens cured at 60 °C for 16 h.

Specimen No.	Flexural Strength (MPa)	Load (N)	Elongation at Break (mm)
P1	65.21	32.03	17.60
P2	61.79	32.32	16.44
P3	60.72	30.66	15.99
P4	67.19	33.10	17.52
P5	60.16	31.50	16.58
Measurement error (%)	±3	±1	±0.5

**Table 6 polymers-12-01052-t006:** Flexural tests for specimens cured at 80 °C for 2 h.

Specimen No.	Flexural Strength (MPa)	Load (N)	Elongation at Break (mm)
P1.1	59.62	30.53	15.97
P2.1	62.55	30.73	15.40
P3.1	59.02	27.16	15.67
P4.1	61.48	29.18	16.63
P5.1	62.21	29.40	15.90
Measurement error (%)	±3	±1	±0.5

**Table 7 polymers-12-01052-t007:** Flexural tests for method 5 specimens.

Specimen No.	Flexural Strength (MPa)	Load (N)	Elongation at Break (mm)
P1	2.33	0.13	0.01
P2	22.99	15.71	14.14
P3	2.26	−0.05	0.04
Measurement error (%)	±3	±1	±0.5

**Table 8 polymers-12-01052-t008:** Method 6 flexural test results for PUF-DCPD specimens.

Specimen No.	Flexural Strength (MPa)	Load (N)	Elongation at Break (mm)
P1	48.07	24.89	15.41
P2	48.31	25.12	15.38
P3	50.24	24.16	16.05
P4	52.40		
P5	43.31	22.60	16.33
Measurement error (%)	±3	±1	±0.5

**Table 9 polymers-12-01052-t009:** Flexural test results for MUF-ENB specimens, cured at 80 °C for 2 h, according to method 5.

Specimen No.	Flexural Strength (MPa)	Load (N)	Elongation at Break (mm)
P1	16.32	9.40	4.91
P2	17.87	10.81	5.34
P3	19.59	10.30	4.66
P4	22.13	10.48	5.08
P5	24.45	16.71	5.00
Measurement error (%)	±3	±1	±0.5

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
