# Peer review of "The Synthetization and Analysis of Dicyclopentadiene and Ethylidene-Norbornene Microcapsule Systems"

_polymers, 2020, doi:10.3390/polym12051052_

Round 1

Reviewer 1 Report

The authors describe the synthesis and analysis of microcapsule systems for self-healing systems. The study is exectued well, however, there are major points, which has to be addressed.

1.) Figure 1 is not required. Instead, an overivew of the different capsule system utilized for self-healing is required.

2.) The stability of the capusle is tested. However, a major problem of the extrinsic self-healing system is the long-term leakage of these. Therefore, long-term leakage of those systems must be analyzed.

3.) Self-healing tests were not performed. These should be performed since the success of the novel approach should be demonstrated.

4.) The paper is too long and should be more concise. Not all figures are required and a SI document should be added.

5.) Major references for extrinsic self-healing systems are not mentioned/cited. Thus, more citations are required.

Author Response

First of all we want to thank you for your time in reviewing the paper and for you inputs. We do hope that the revised version is more suitable to you, even though it is still longer than expected. Below, you can find our responses point-by-point to your review. The paper is re-submitted with Track Changes.

1.) Figure 1 is not required. Instead, an overview of the different capsule system utilized for self-healing is required.

As part of the introduction, we considered to keep the first Figure. In addition, a more detailed research has been made and a list of different capsule system has been foreseen. This can be seen at Line 73.

2.) The stability of the capsule is tested. However, a major problem of the extrinsic self-healing system is the long-term leakage of these. Therefore, long-term leakage of those systems must be analyzed.

Indeed, you are right, the long-term leakage of the presented self-healing system must be analyzed, but this was not the purpose of our study. However, a starting point for the long-term leakage analysis was conducted in Reference 33, after thermal cycling from -20 dgr.C to +100dgr.C for 12 hours. We are indeed considering another study that we will analyze the long-term leakage of such systems.

3.) Self-healing tests were not performed. These should be performed since the success of the novel approach should be demonstrated.

Initially, we didn’t consider presenting the self-healing results for the proposed method, as they were meant for another paper. However, following your review, we understood its importance in understanding the results. The self-healing tests and evaluation are presented starting from Line 485.

4.) The paper is too long and should be more concise. Not all figures are required and a SI document should be added.

Indeed, the paper is longer than expected, but we consider that the content is needed for understanding the methodology and may help in further research. However, a revision of the paper has been made, also in accordance with Reviewer No.2. All modifications are made with track changes and hopefully the information is more clearly now.                                               

5.) Major references for extrinsic self-healing systems are not mentioned/cited. Thus, more citations are required.

Considering your first review point, we did a more detailed research and several other citations have been added emphasizing the evaluation of extrinsic self-healing on polymers and polymer composites.

Reviewer 2 Report

The paper discusses the fabrication and characterization of two microcapsules systems. The research direction is prevalent these days. However, the manuscript can not be published in its present form. The presentation and discussion of the results must be improved. The structure of the paper is too confusing, the experimental section showing results, and the result section elaborating on the testing procedure. The authors are suggested to revise the structure of the manuscript on resubmission.

As much as I prefer to go straight to the testing itself, in this paper, it will be better to have a bit more detail in the introduction with previous literature. The authors are advised to follow standard scientific writing practices, for instance, the authors stated "Several researchers" accompanied by a single citation. Please revise it.

Table 3 refers to six methods in roman numerals, and later in the manuscript, they are cited as Arabic numerals.

Table 3 Method no. VI, than => then

Please add the measurement error of all individual measurements in Tables 4 to 8.

Page 12, line 280. Please give other justification since there are already well-established methods.

How the authors are discussing "ultrasound stirrer ... microcapsules dimensions". Scientific writing is not about ideas and beliefs; instead, there must be an authentic reference or experimental results.

Where are the tests showing self-healing?

The conclusion is poorly written. The authors have not summed up the results in a concise way.

Merge Figures 2 to 4 into one eye-catching scheme.

Figure 12, The y-axis and peaks marked are not clear to readers.

Figure 13, TGA curve, please check the location of -38.707%

Figure 15, Mark for what these two images refer to

Figure 16 - too basic to be added here.

Check Figure caption of Figure 17; it seems misleading

Figures 21 and 22 are confusing; they are displaying two coupons. The figure caption does not mention which pictures belong to what? Mainly, in Figure 22, samples are showing 3 and 4 written over them, they further confuse where were the 1 and 2.

Figure 22 is related to section 3.1.2. Method 2. It pretends its relation with the following section 3.1.3., referring to Method 3 and Method 4 since the coupons were not displayed for later sections.

What is the purpose of Figure 23 in this manuscript?

There are several typos throughout the manuscript and grammatical errors. For example, Page 18, line 410 three point bending => three-point bending.

Check the sentence fragment on Page 16, Line 383: Fibre-reinforced plastic composites.

Punctuation marks are also missing.

Besides, the manuscript is full of inconsistencies. It seems the authors have pasted big chunks of the sentence as they were copied. For instance, page 22, line 478, 2 hours vs. 16h

Page 5 line 128 cross-linked  vs. Page 15, line 345, crosslinked

Author Response

First of all we want to thank you for your time in reviewing the paper and for you inputs. We do hope that the revised version is more suitable to you, even though it is still longer than expected. Below, you can find our responses point-by-point to your review. The paper is re-submitted with Track Changes.

1) As much as I prefer to go straight to the testing itself, in this paper, it will be better to have a bit more detail in the introduction with previous literature. The authors are advised to follow standard scientific writing practices, for instance, the authors stated "Several researchers" accompanied by a single citation. Please revise it.

We made the modification accordingly. Please see Line 46, Line 47 and Line 64.

2) Table 3 refers to six methods in roman numerals, and later in the manuscript, they are cited as Arabic numerals.

Modifications have been made according to your remark and can be found at Line 353. Also, we added another Table in the Introduction Section, thus Table 3 is now Table 4.

3) Table 3 Method no. VI, than => then

Indeed, it was a typo. Thank you for noticing.

4) Please add the measurement error of all individual measurements in Tables 4 to 8.

For Tables 5 to 9 a row was added indicating the measurement errors

5) Page 12, line 280. Please give other justification since there are already well-established methods.

An appropriate justification is made at Line 278. The justification mentioned in Line 306 (280 before) was deleted.

6) How the authors are discussing "ultrasound stirrer ... microcapsules dimensions". Scientific writing is not about ideas and beliefs; instead, there must be an authentic reference or experimental results.

Discussions Section was revised and summarized for a better understanding

7) Where are the tests showing self-healing?

Initially, we didn’t consider presenting the self-healing results for the proposed method, as they were meant for another paper. However, following your review, we understood its importance in understanding the results. The self-healing tests and evaluation are presented starting from Line 485.

8) The conclusion is poorly written. The authors have not summed up the results in a concise way.

Conclusion Section has been modified. We hope that the revised version is more concise

9) Merge Figures 2 to 4 into one eye-catching scheme.

Figures 2 to 4 were combined into a single figure as a schematic description of the process is presented in Figure 3

10) Figure 12, The y-axis and peaks marked are not clear to readers.

Figure 11 (Figure 12 before) presenting the FT-IT spectra has been modified accordingly.

11) Figure 13, TGA curve, please check the location of -38.707%

Figure 12 (Figure 13 before) has been modified accordingly

12) Figure 15, Mark for what these two images refer to

Figure 14 (Figure 15 before) has been modified accordingly

13) Figure 16 - too basic to be added here.

14) Check Figure caption of Figure 17; it seems misleading

With respect to Points 13 and 14. For a better understanding, we deleted Figure 16b (Figure 17b before) am replaced it with Figure 15 (Figure 16 before).

15) Figures 21 and 22 are confusing; they are displaying two coupons. The figure caption does not mention which pictures belong to what? Mainly, in Figure 22, samples are showing 3 and 4 written over them, they further confuse where were the 1 and 2.

16) Figure 22 is related to section 3.1.2. Method 2. It pretends its relation with the following section 3.1.3., referring to Method 3 and Method 4 since the coupons were not displayed for later sections.

With respect to Points 15 and 16. We merged the two figures in one figure (Figure 19) and moved them at Line 372. The caption was also modified accordingly. Also, we changed the two samples having 3 and 4 written over them, to avoid any further confusions. As a remark, these two samples were taken out from a batch of five samples fabricated using Method 2.

Moreover, Figure 20 was added presenting both PUF-DCPD and MUF-ENB samples fabricated by Method 6

17) What is the purpose of Figure 23 in this manuscript?

It was just an exemplification of fabricated samples.  However, considering your review point, and by adding Figure 20, we consider to delete this figure.

18) There are several typos throughout the manuscript and grammatical errors. For example, Page 18, line 410 three point bending => three-point bending.

19) Check the sentence fragment on Page 16, Line 383: Fibre-reinforced plastic composites.

20) Punctuation marks are also missing.

21) Besides, the manuscript is full of inconsistencies. It seems the authors have pasted big chunks of the sentence as they were copied. For instance, page 22, line 478, 2 hours vs. 16h

Response for Points 18 to 20. The paper was revised for inconsistencies and modifications were done accordingly. With respect to Point 20, indeed, some part of the text was taken from a progress report on an ongoing project but it was revised.

22) Page 5 line 128 cross-linked  vs. Page 15, line 345, crosslinked

Thank you for your observation. The document was revised and adjustments have been made.

Round 2

Reviewer 1 Report

The authors worked intensively on the manuscript and improved it in a suitable manner. Therefore, I suggest publication in the current form.

Reviewer 2 Report

The authors have improved manuscript significantly, as per the comments of the reviewers, therefore, this manuscript can be considered in its present form.